# Multi-Lingual Acquisition on Multimodal Pre-training for Cross-modal Retrieval

**Liang Zhang[1], Anwen Hu[1], Qin Jin[1,2,∗]**
[1]School of Information, Renmin University of China
[2]Key Laboratory of Data Engineering and Knowledge Engineering (MOE),
Renmin University of China
{zhangliang00,anwenhu,qjin}@ruc.edu.cn

## Abstract

Vision and diverse languages are important information sources in our living world. A model that understands multi-modalities and multi-languages can be applied to a wider range of real-life scenarios. To build such a multimodal and multilingual model, existing works try to ensemble vision-language data from multiple languages in pre-training. However, due to the large number of languages, these works often require huge computing resources and cannot be flexibly extended to new languages. In this work, we propose a **M**ulti-**L**ingual **A**cquisition (MLA) framework that can easily empower a monolingual Vision-Language Pre-training (VLP) model with multilingual capability. Specifically, we design a lightweight language acquisition encoder based on state-of-the-art monolingual VLP models. We further propose a two-stage training strategy to optimize the language acquisition encoder, namely the Native Language Transfer stage and the Language Exposure stage. With much less multilingual training data and computing resources, our model achieves state-of-the-art performance on multilingual image-text and video-text retrieval benchmarks.

## 1   Introduction

We live in a multimodal and multilingual world. The information we receive in our daily lives may come from different modalities and languages. Therefore, building multimodal and multilingual models to effectively understand such information has attracted much research attention [12, 38, 21, 3]. Recently, Multilingual Vision-Language Pre-training (M-VLP) achieves convincing performance in various cross-lingual cross-modal tasks such as multilingual image-text retrieval [28, 44, 11, 16, 18] and multimodal machine translation [35]. As shown in Figure 1(a), M-VLP models handle multiple

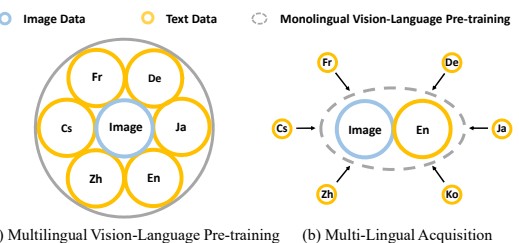

(a) Multilingual Vision-Language Pre-training    (b) Multi-Lingual Acquisition

Figure 1: Comparison of data usage between M-VLP and MLA. The size of a circle reflects the amount of training data.

languages and modalities simultaneously during pre-training. Despite their successes, M-VLP models suffer from two problems. First, pre-training on vision and multilingual data consumes huge computing resources. For example, the state-of-the-art M-VLP model MURAL [18] is pre-trained on 128 Cloud TPUv3 for four days. It could support multimodal tasks on 100+ languages. However, considering there are 6,900+ languages worldwide [44], building such a single model to handle all

---

∗Corresponding author

36th Conference on Neural Information Processing Systems (NeurIPS 2022).

languages will be highly expensive. Second, M-VLP models cannot be flexibly extended to new languages. Additional training is required for M-VLP models to achieve satisfactory performance on a new language. However, this training process will cause performance degeneration of M-VLP models on the original languages due to the limited model capacity. For example, the limited model capacity even results in M-VLP models performing worse than their monolingual counterparts on English [28, 44].

To build multimodal and multilingual models with low-cost and high-flexibility, we refer to our human learning habits when acquiring new languages. We humans normally learn our native language during childhood and practice it through interactions with the multimodal living environments. When learning a new language, we humans initially tend to align it with the native language, as we can easily map words in the native language to real-world objects and concepts. After having a certain language foundation, we could further master it by interacting with the environment directly using the new language. This is known as the language exposure [5]. The whole learning process rarely degrades our native language capability.

Inspired by this, we propose a new framework, **M**ulti-**L**ingual **A**cquisition (MLA), which constructs multimodal and multilingual models based on monolingual VLPs. The topology of the MLA-based multimodal and multilingual model is illustrated in Figure 1(b). Unlike M-VLPs, which handle data from multiple languages and modalities in a single model, MLA empowers monolingual VLPs with multilingual capability using much less training data through a language acquisition encoder. The language acquisition encoder is realized by inserting our proposed lightweight language acquirers into the pre-trained monolingual encoder of the VLP model. During training, original parameters in the pre-trained monolingual encoder are fixed, only multi-lingual embeddings and language acquirers for each new language are optimized. Following the human learning habits, we propose a two-stage training strategy to train the language acquisition encoder. In the Native Language Transfer (NLT) stage, the model is trained to establish the correspondence between the new languages with the native language. In the Language Exposure (LE) stage, the model is optimized to build cross-modal alignment between new languages and images. We apply our proposed MLA to the monolingual VLP model CLIP [31] and achieve state-of-the-art results on both multilingual image-text and video-text retrieval benchmarks with much less training data and computing resources. Ablation studies demonstrate the effectiveness of our training strategy. Owing to the independence merit of the language acquirers, the MLA-based models can be easily extended to new languages without compromising the performance of their original languages. Since cross-modal retrieval is a fundamental task for multilingual and multimodal learning, and most M-VLP [28, 44, 18] works are evaluated using this task, we focus on cross-modal retrieval in the paper. The application of MLA to other tasks will be explored in our future work.

The main contributions of our work are as follows: 1) We propose a lightweight Multi-Lingual Acquisition (MLA) framework that can easily empower monolingual VLPs with multilingual capability. 2) We propose a two-stage training strategy to optimize the MLA-based models inspired by the language learning habits of humans. Ablation studies prove the effectiveness of the strategy. 3) We apply MLA to the monolingual VLP model CLIP and achieve the new state-of-the-art results on both multilingual image-text and video-text retrieval benchmarks with much less training data and parameters.

## 2   Related Work

**Vision-Language Pre-training:** There are increasing interest in building Vision-Language Pre-training (VLP) models. From the perspective of how to interact between vision and language modalities, existing models can be divided into two categories: single-stream and dual-stream models. The single-stream models perform interaction on image and text directly with a cross-modal transformer [7, 26, 22]. In contrast, the dual-stream models encode image and text with two independent encoders and optimize via simple objectives like image-text contrastive learning [31, 19, 43]. Compared with the single-stream models, the dual-stream models are more efficient to utilize noisy image-text data harvested from the web [17], and thus achieve better performance on downstream tasks. Meanwhile, the dual-stream models are more flexible for extension. Since the dual-stream models process images and text through independent encoders, we can fix the vision encoders and focus on extending the text encoders to support new languages. Therefore, we focus on empowering dual-stream VLPs with multilingual capability in this work.

**Multilingual Vision-Language Pre-training:** To achieve both multilingual and multimodal capability, many works try to learn the relationship between multiple languages and modalities simultaneously through pre-training. M$^3$P [28] introduces the multimodal code-switched training method to enhance multilingual transferability. UC$^2$ [44] augments the English image-text data to other languages through machine translation and proposes fine-grained pre-training objectives to encourage alignment between image regions and multilingual tokens. More recently, MURAL [18] adopts the dual-stream structure. It is pre-trained with image-text and text-text contrastive objectives on multilingual image-text pairs and translation pairs. M-VLP models significantly outperform previous non-pretraining models [12, 38, 21, 3] on multilingual image-text retrieval. Despite their success, these models typically consume huge computing resources and large-scale multilingual training data. Moreover, they fail to take full advantage of the cross-modal knowledge learnt in monolingual VLP, and building cross-modal cross-lingual representations from scratch can be very hard. In contrast, our MLA framework aims to empower VLP models with multilingual capability and it builds multimodal and multilingual models with much less data and computing cost.

**Multilingual Extension:** Some works explore making pre-trained monolingual language models multilingual [32, 30, 36]. Reimers et al. [32] extend sentence embeddings from monolingual to multilingual by Multilingual Knowledge Distillation (MKD). Artetxe et al. [2] extend monolingual models by training additional word embeddings . MAD-X [30] extends multilingual pre-training models to support low-resource languages through adapters [15]. By extending state-of-the-art pre-trained language models, these works have achieved impressive results in NLP tasks such as bitext retrieval [32], cross-lingual QA and NER [30, 32]. However, few works focus on making VLP models multilingual. OSCAR+$^{Ada}$ [29] extends single-stream model OSCAR [26] to multilingual for VQA tasks by adopting a similar strategy with MAD-X [30]. It trains language adapters with Masked Language Modeling (MLM) for each language, and replaces the English adapter with the target language one during inference. This strategy [30, 29] shares the same idea with MLA that handles different languages through independent structures but with two core differences: i) It is designed for MLM-based VLPs and cannot be applied to dual-stream models such as CLIP [31]. Consequently, it is not suitable for cross-modal retrieval considering the gap between retrieval and MLM. ii) It generalizes poorly on other languages since MLM can only implicitly establish the correspondence between these languages and English, let alone vision correspondences. In contrast, MLA directly builds the connection between the other languages with English and then with vision through the two-stage training strategy. It thus achieves comparable results on these languages as on English in the downstream retrieval tasks.

## 3 Method

The Multi-Lingual Acquisition (MLA) framework is proposed to empower a dual-stream monolingual VLP model with multilingual capability. We define the *native language* of a VLP as its pre-training language. In this paper, we choose CLIP-ViT-B [31] as the VLP model. It is pre-trained with 400M image-text pairs in English [31]. Note that MLA can also be applied to VLP models with different native languages.

Since the state-of-the-art VLP models can project vision and native language into a shared multimodal space, we design a language acquisition encoder to process non-native languages. We then simulate the learning habits of human beings and propose a two-stage training strategy to optimize the language acquisition encoder. We first introduce the architecture of the MLA framework in Sec.3.1. Then, we describe our training strategy in Sec.3.2.

### 3.1 Architecture

Figure 2(a) illustrates the overview of the MLA framework, which consists of three modules: the pre-trained text encoder, the pre-trained vision encoder, and the language acquisition encoder.

**Pre-trained Text Encoder.** Given a sentence $S$ in the native language, the corresponding sentence representation $s = \Phi(S; \theta_\Phi)$ is generated through the pre-trained text encoder $\Phi$. To preserve the cross-model knowledge of VLP, $\theta_\Phi$ is keep fixed during training. As shown in the top part of Figure 2(a), the pre-trained text encoder contains a native embedding block and $l$ transformer layers [37]. The native embedding block first tokenizes $S$ with byte pair encoding (BPE) [33]. Then, it converts words into embeddings $E_S = [e_{0=\texttt{[SOS]}}, e_1, \ldots, e_{M=\texttt{[EOS]}}]$. $\texttt{[SOS]}$ and $\texttt{[EOS]}$ are special

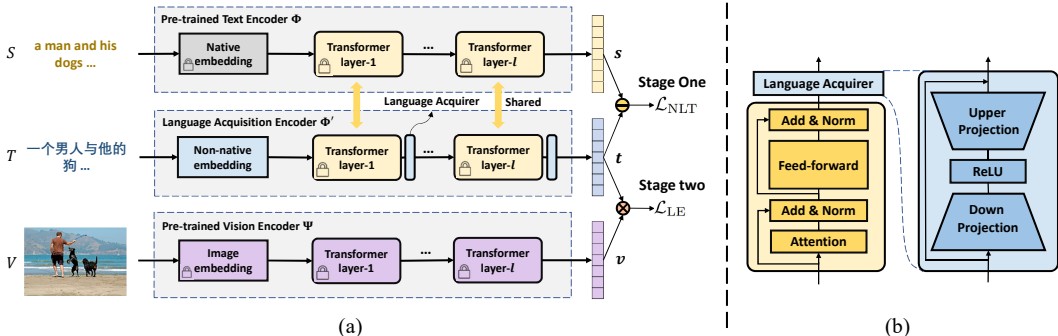

Figure 2: Model illustration: (a) The overview of MLA framework. (b) The structure of a language acquirer

tokens denoting the boundary of $S$. The word embeddings are then passed through the transformer layers:

$$H^0 = [e_{0=\texttt{[SOS]}}, e_1, \ldots, e_{M=\texttt{[EOS]}}] + E_{pos} \tag{1}$$

$$H^i = \texttt{TransformerLayer}(H^{i-1}; \theta_\Phi^i) \tag{2}$$

where $H^i = [h_0^i, \ldots, h_M^i]$ is the hidden state of the layer $i$. $\theta_\Phi^i$ denotes the parameters of the layer $i$. $E_{pos}$ is the positional encoding. Note that the causal self-attention mask is used in the transformer layers [31]. The last hidden state of the [EOS] token is chosen to generate the sentence representation:

$$\boldsymbol{s} = W_a h_M^l \tag{3}$$

where $\boldsymbol{s}$ is the sentence representation of $S$, and $W_a$ denotes a linear projection.

**Pre-trained Vision Encoder.** We extract the representation $\boldsymbol{v} = \Psi(V; \theta_\Psi)$ of an image $V$ with the pre-trained vision encoder $\Psi$. Similar with the pre-trained text encoder, $\theta_\Psi$ is also frozen. The pre-trained vision encoder is implemented as a Vision Transformer [9]. As shown in the bottom part of Figure 2(a), it consists of a image embedding block and $l$ transformer layers. Given an image $V$, the image embedding block first divides $V$ into patches $V' = [v_1', \ldots, v_N']$ following [9]. Then, they are linearly projected into patch embeddings $E_p = [e_{\texttt{[CLASS]}}, W_p v_1', \ldots, W_p v_N']$, where $e_{\texttt{[CLASS]}}$ is a special embedding for the whole image and $W_p$ is the linear projection. The patch embeddings are then fed into transformer layers:

$$Z^0 = [e_{\texttt{[CLASS]}}, W_p v_1', \ldots, W_p v_N'] + E_{pos} \tag{4}$$

$$Z^i = \texttt{TransformerLayer}(Z_{i-1}; \theta_V^i) \tag{5}$$

where $Z^i = [z_0^i, \ldots, z_N^i]$ is the hidden state of the layer $i$. The last hidden state of the [CLASS] embedding $z_0^l$ is selected to produce the representation of image $V$:

$$\boldsymbol{v} = W_b z_0^l \tag{6}$$

where $\boldsymbol{v}$ is the image representation of $V$, and $W_b$ denotes a linear projection.

**Language Acquisition Encoder.** As shown in the middle part of Figure 2(a), the language acquisition encoder is built upon the pre-trained text encoder. Suppose $T$ is a sentence written in a non-native language $L$, we get the representation of $T$ through language acquisition encoder $\boldsymbol{t} = \Phi'(T; \theta_\Phi, \theta_{emb}, \theta_L)$, where $\theta_\Phi$ are fixed parameters of the pre-trained text encoder, $\theta_{emb}$ refers to a shared non-native embedding block and $\theta_L$ represents specialized language acquirers for language $L$. Non-native sentence $T$ is first tokenized and processed into word embeddings $E_T = [u_{0=\texttt{[SOS]}}, \ldots, u_{M=\texttt{[EOS]}}]$ through the non-native embedding block. The word embeddings are then encoded through the pre-trained transformer layers and language acquirers:

$$X^0 = [W_e u_{0=\texttt{[SOS]}}, W_e u_1, \ldots, W_e u_{m=\texttt{[EOS]}}] + E_{pos} \tag{7}$$

$$H^i = \texttt{TransformerLayer}(X^{i-1}; \theta_\Phi^i) \tag{8}$$

$$X^i = \texttt{LA}(H^i; \theta_L^i) \tag{9}$$

where $X^i = [x^i_0, \ldots, x^i_m]$ is the hidden state of the layer $i$. $W_e$ is a linear projection to keep dimension consistency. $\theta^i_L$ denotes the parameters of the $i$-th language acquirer for language $L$. As shown in Figure 2(b), the language acquirer is implemented as a bottleneck MLP with residual connection [14]:

$$\mathtt{LA}(X) = W_{upper}\mathtt{ReLU}(W_{down}X) + X \tag{10}$$

Similar with the pre-trained text encoder, the last hidden state of the [EOS] token is projected into the sentence representation $\boldsymbol{t}$:

$$\boldsymbol{t} = W_a x^l_m \tag{11}$$

Eq.11 shares the same linear projection $W_a$ with Eq.3. The main advantage of the language acquisition encoder is that it can extend the VLP models to support new languages without influencing the existing languages, as it handles different languages with independent language acquirers.

## 3.2 Training Strategy

To simulate the language learning habits of humans, we optimize the model in two stages: the Native Language Transfer (NLT) stage and the Language Exposure (LE) stage.

**Native Language Transfer.** When learning a new language, we humans initially tend to align it with the native language. To simulate this learning phase, we align the non-native representations to the native representations during the Native Language Transfer (NLT) stage. Specifically, suppose $\{(S_1, T_1), ..., (S_n, T_n)\}$ are translation pairs, where $S_i$ is in the native language, and $T_i$ is in a non-native language $L$. The objective in the NLT stage is minimizing the Mean Square Error (MSE) between the native representation $\boldsymbol{s}_i = \Phi(S_i; \theta_\Phi)$ and the non-native representation $\boldsymbol{t}_i = \Phi'(T_i; \theta_\Phi, \theta_L, \theta_{emb})$:

$$\mathcal{L}_{\mathrm{NLT}} = \frac{1}{B}\sum_{i=1}^{B}\|\boldsymbol{s}_i - \boldsymbol{t}_i\|^2 \tag{12}$$

where B is the batch size. Note that $\theta_\Phi$ is loaded from the VLP model and is kept frozen. $\theta_L$ is trained for non-native language $L$. $\theta_{emb}$ is shared among non-native languages.

During the NLT stage, the non-native language correspondence with vision can be built by pivoting on the native language, since the correspondence between the native language and vision is well established through VLP.

**Language Exposure.** After the NLT stage, the model has built an implicit connection between non-native languages and vision. However, due to the existence of synonyms, two same words in the native language may correspond to different images. Thus, ambiguity may arise when learning non-native languages solely by relying on the native language. Actually, we can regard the language acquisition encoder after the NLT stage as a person with a certain language foundation. He/She has learned the basic usage of a language through native language teaching. To master it, he/she may practice the non-native language by interacting with the multimodal living environments. Inspired by this learning phase, we directly establish the cross-modal alignment between non-native languages and vision during the Language Exposure (LE) stage. Given image-text pairs $\{(V_1, T_1), ..., (V_n, T_n)\}$ where $T_i$ is in a non-native language $L$, the sentence representation $\boldsymbol{t}_i = \Phi'(T_i; \theta_\Phi, \theta_L, \theta_{emb})$ should be closer to the aligned image representation $\boldsymbol{v}_i = \Psi(V_i; \theta_\Psi)$, and away from the misaligned one $\boldsymbol{v}_j = \Psi(V_j; \theta_\Psi), j \neq i$. This can be achieved by performing contrastive learning between non-native languages and images. For a non-native sentence $T_i$, we treat the corresponding image $V_i$ as a positive sample, and other images in the same batch $V_j, j \neq i$ as negative samples. Vice versa for images. The objective in the LE stage is minimizing the NCE loss [13] defined as follows:

$$\mathcal{L}_{\mathrm{LE}} = \frac{1}{2}(\mathcal{L}_{v2t} + \mathcal{L}_{t2v}) \tag{13}$$

$$\mathcal{L}_{v2t} = -\frac{1}{B}\sum_{i=1}^{B}\log\frac{\exp(\mathrm{sim}(\boldsymbol{v}_i, \boldsymbol{t}_i)/\tau)}{\sum_{k=1}^{N}\exp(\mathrm{sim}(\boldsymbol{v}_i, \boldsymbol{t}_k)/\tau)} \tag{14}$$

$$\mathcal{L}_{t2v} = -\frac{1}{B}\sum_{i=1}^{B}\log\frac{\exp(\mathrm{sim}(\boldsymbol{v}_i, \boldsymbol{t}_i)/\tau)}{\sum_{k=1}^{N}\exp(\mathrm{sim}(\boldsymbol{v}_k, \boldsymbol{t}_i)/\tau)} \tag{15}$$

where $B$ is the batch size. $\text{sim}(\boldsymbol{x}, \boldsymbol{y}) = \frac{\boldsymbol{x}^\top \boldsymbol{y}}{\|\boldsymbol{x}\|\|\boldsymbol{y}\|}$ is the cosine similarity between two vectors. $\tau$ is a temperature hyper-parameter to scale the logits. Note that though the image-to-text loss $\mathcal{L}_{v2t}$ is optimized, the pre-trained vision encoder is kept frozen during training. Similar to NLT, the trainable parameters in LE come from the language acquirers and the non-native embedding block.

## 4   Experiments

### 4.1   Dataset Description

We train our model with the Conceptual Captions (CC) dataset [34] and two translation enhanced versions of the CC [44, 4]. We use Multi30K [10], MSCOCO [6, 25, 40] and XTD [1] for multilingual image-text retrieval evaluation, and MSRVTT [39, 16] for multilingual video-text retrieval evaluation. **Conceptual Captions** (CC) [34] contains 3.3 million image-text pairs in English crawled from the Web.[2] We also randomly select 300K image-text pairs denoted as **CC300K** for training our model to show the low-cost merit of MLA. For multilingual sentences, we leverage two translation augmented CC datasets: (1) **CC6L** [44] that translates all English captions of the CC into 5 languages (German(de), French(fr), Czech(cs), Chinese(zh));[3] and (2) **CC69L** [4] that contains 27K captions in each of the 68 languages translated from English.[4] Considering the languages of the downstream datasets, we train the model with CC6L for multilingual image-text retrieval, and with CC69L for multilingual video-text retrieval.
**Multi30K** [10] is built upon Flickr30K [41]. The English(en) captions are manually translated into German(de), French(fr) and Czech(cs). It contains 31K images paired with 5 captions per image in English and German, and 1 caption in French and Czech. We use the standard train, dev and test splits defined by Young et al. [41].
**MSCOCO** [6] contains 123K images with 5 English captions per image. Yoshikawa et al. [40] annotate 5 Japanese captions per image, and Li et al. [25] extends MSCOCO with Chinese captions for 20K images. We follow the standard train, dev and test splits for English and Japanese as in [20]. For Chinese, we can only perform zero-shot evaluation on the test split defined by Li et al. [25], as the full splits have overlaps with English and Japanese splits.
**XTD** [1] provides captions in 11 languages (English(en), German(de), French(fr), Chinese(zh), Japanese(ja), Italian(it), Spanish(es), Russian(ru), Polish(pl), Turkish(tr), Korean(ko)) for 1K MSCOCO images. Except for Japanese, all non-English captions are translated from the English caption directly. We use this dataset for zero-shot image-text retrieval evaluation only.
**MSRVTT** [39] is a video caption dataset with 10K videos, where each video is annotated with 20 English captions. Huang et al.[16] translates the English captions into 8 languages (German(de), French(fr), Russian(ru), Spanish(es), Czech(cz), Swahili(sw), Chinese(zh) and Vietnamese(vi)) via machine translation service. We follow the standard train/dev splits in [39], and evaluate on the 1K test split as described in [42].

### 4.2   Implementation Details

We apply MLA on two VLP models: CLIP-ViT-B-32 and CLIP-ViT-B-16 [31], denoted as $\text{MLA}_{\text{CLIP}}$ and $\text{MLA}_{\text{CLIP16}}$ respectively. The hidden dimension of the language acquirers is set to 256, and all language acquirers for each non-native language cost only 3.14 MB parameters. The non-native embedding matrix is initialized with M-BERT [8]. It costs 92.2 MB and shared with all non-native languages. We train two separate models for multilingual image-text retrieval and video-text retrieval. For the image model, we train with CC6L [44]. For the video model, we use multilingual captions from CC69L [4]. For both models, we optimize multiple language acquirers iteratively with a batch size of 128. The NLT stage performs 117,150 steps with a learning rate of 1e-4, and the LE stage performs 11,715 steps with a learning rate of 3e-6. The temperature $\tau$ is set to 0.01. For both stages, we use the Adam optimizer [23] with a linear warm-up for the first 10% of steps. The whole training process takes about 12 hours to converge on 1 Nvidia V100 GPU.

---

[2]We can only access $\sim$2.5 million images due to some broken URLs.
[3]Dataset released at `https://github.com/zmykevin/UC2`, under MIT license.
[4]Released at `https://github.com/FreddeFrallan/Multilingual-CLIP`, under MIT license. We remove captions of unaccessible images, leaving $\sim$20K captions for each language.

Table 1: Multilingual image-text retrieval results on Multi30K and MSCOCO. TrTrain: Translate-train, FT-En: *Fine-tune on English*, FT-All: *Fine-tune on All*. ◇: Models re-implemented by us. †: Models trained with publicly unavailable datasets. ‡: Models fine-tuned on COCO-CN [25], which has an overlap train split with the test split of English and Japanese. Best results are in **bolded** and second best are underlined.

| | Method | Training Data | Multi30K | | | | MSCOCO 1K | | MSCOCO 5K | |
|---|---|---|---|---|---|---|---|---|---|---|
| | | | en | de | fr | cs | en | ja | en | ja |
| Zero-shot | Unicoder-VL | CC3M (English only) | 72.0 | - | - | - | 63.7 | - | - | - |
| | ALIGN | AT-en (English only) | 84.3 | - | - | - | 80.0 | - | 60.6 | - |
| | M³P | CC3M+Wiki | 57.9 | 36.8 | 27.1 | 20.4 | 63.1 | 33.3 | - | - |
| | UC² | TrTrain(CC3M) | 66.6 | 62.5 | 60.4 | 55.1 | 70.9 | 62.3 | - | - |
| | MKD$_{CLIP}$ | TrTrain(CC300K) | 82.1 | 77.1 | 75.2 | 72.3 | 78.5 | 73.6 | - | - |
| | MURAL◇ | TrTrain(CC300K) | 67.8 | 62.7 | 60.8 | 57.5 | 68.1 | 62.5 | 43.3 | 37.1 |
| | MURAL | TrTrain(CC12M)+EOBT | 80.9 | 76.0 | 75.7 | 68.2 | 78.1 | 72.5 | 58.0 | 49.7 |
| | MURAL† | AT+MBT | 82.4 | 76.2 | 75.0 | 64.6 | 79.2 | 73.4 | 59.5 | 54.4 |
| | MLA$_{CLIP}$ | TrTrain(CC300K) | 84.4 | 78.7 | 77.7 | 70.8 | 79.4 | 74.9 | 60.5 | 54.1 |
| | MLA$_{CLIP16}$ | TrTrain(CC300K) | **86.4** | **80.8** | **80.9** | **72.9** | **80.9** | **76.7** | **62.6** | **57.0** |
| FT-En | M³P | CC3M+Wiki | 87.4 | 82.1 | 67.3 | 65.0 | 88.6 | 56.0 | - | - |
| | UC² | TrTrain(CC3M) | 87.2 | 83.8 | 77.6 | 74.2 | 88.1 | 71.7 | - | - |
| | MURAL◇ | TrTrain(CC300K) | 82.8 | 76.5 | 76.7 | 70.1 | 84.5 | 74.6 | 64.3 | 52.5 |
| | MLA$_{CLIP}$ | TrTrain(CC300K) | 92.0 | 82.6 | 85.1 | 76.2 | 89.3 | 80.4 | 75.7 | 62.1 |
| | MLA$_{CLIP16}$ | TrTrain(CC300K) | **94.5** | **86.4** | **87.3** | **79.5** | **91.3** | **82.6** | **79.4** | **65.5** |
| FT-All | M³P‡ | CC3M+Wiki | 87.7 | 82.7 | 73.9 | 72.2 | 88.7‡ | 87.9‡ | - | - |
| | UC²‡ | TrTrain(CC3M) | 88.2 | 84.5 | 83.9 | 81.2 | 88.1‡ | 87.5‡ | - | - |
| | MURAL◇ | TrTrain(CC300K) | 83.4 | 80.3 | 79.4 | 76.8 | 84.8 | 83.2 | 64.7 | 62.9 |
| | MURAL | TrTrain(CC12M)+EOBT | 91.0 | 87.3 | 86.4 | 82.4 | 89.4 | 87.4 | 73.7 | 71.9 |
| | MURAL† | AT+MBT | 92.2 | 88.6 | 87.6 | 84.2 | 88.6 | 88.4 | 75.4 | 74.9 |
| | MLA$_{CLIP}$ | TrTrain(CC300K) | 92.0 | 86.8 | 85.4 | 82.3 | 89.3 | 88.1 | 75.7 | 73.2 |
| | MLA$_{CLIP16}$ | TrTrain(CC300K) | **94.5** | **89.7** | **89.2** | **85.9** | **91.3** | **90.4** | **79.4** | **76.5** |

## 4.3 Evaluation on Multilingual Image-Text Retrieval

In multilingual image-text retrieval, models are given a sentence in a certain language to find the most semantically relevant image from an image database and vice versa. We compare our model with state-of-the-art multilingual vision-language pre-training methods under three settings: (1) *Zero-shot:* we directly evaluate the model without fine-tuning on downstream datasets. (2) *Fine-tune on English:* we first fine-tune the VLP model on downstream English data. We then insert the language acquirers and non-native embedding block into the fine-tuned model and evaluate on other languages directly. (3) *Fine-tune on All:* after (2), we fine-tune the language acquirers and non-native embedding block on the downstream dataset and freeze other parts of the model. Following previous works [28, 44, 18], we report Average Recall (AR), which is the average score over Recall@1, Recall@5, and Recall@10 on two retrieval directions (image→text, text→image). The results are shown in Table 1. For a fair comparison, we re-implement MURAL [18] (marked with ◇ in tables) and train it with the same data as MLA.

Under the ***Zero-shot*** setting, we observe that MLA$_{CLIP}$ performs significantly better than state-of-the-art M-VLP models on English. This is because MLA$_{CLIP}$ could completely maintain the strong English performance of CLIP. In contrast, M-VLP models typically perform worse than their monolingual counterparts on English (M³P 57.9 vs. Unicoder-VL [24] 72.0,

Table 2: Comparison over trainable parameters (#TP) and computing costs (Costs).

| Method | Langs | #TP | Costs |
|---|---|---|---|
| M³P | 100 | 566 M | 8×V100×7d |
| UC² | 6 | 478 M | 8×V100×4d |
| MURAL | 109 | 300 M | 128×TPUv3×4d |
| MLA$_{CLIP}$ | 6 | 108 M | 1×V100×0.5d |

MURAL 80.9 vs. ALIGN [19] 84.3). MLA$_{CLIP}$ also outperforms M-VLP models on other languages. For example, MLA$_{CLIP}$ achieves 78.7 average recall on German by using 300K image-text pairs, outperforming MURAL pre-trained with the same data by 16.0% and CC12M+EOBT by 2.7%. It demonstrates that MLA is a high-data-efficient method to empower monolingual VLP models with multilingual capability. Meanwhile, M-VLP models like MURAL require pre-training over a large amount of data to achieve convincing performance. Under the ***Fine-tune on English*** setting, MLA shows strong cross-lingual transfer capability. Under the ***Fine-tune on All*** setting, MLA$_{CLIP}$ performs slightly worse than MURAL which was pre-trained on publicly unavailable dataset AT+MBT [18]. We consider the reason is that MURAL has a larger transformer-based text

encoder (768 hidden dimensions, 12 heads) compared to MLA$_{\text{CLIP}}$ (512 hidden dimensions, 8 heads). This makes it easier for MURAL to fit the downstream datasets with a certain scale such as Multi30K and MSCOCO. MLA$_{\text{CLIP16}}$ achieves state-of-the-art results on all languages under three settings. These results suggest that MLA could probably reach better performance on multilingual image-text retrieval if stronger VLP models such as ALIGN-L2 [19] or Florence [43] are provided.

## 4.4 Evaluation on Multilingual Video-Text Retrieval

In multilingual video-text retrieval, the model searches for the most semantically relevant videos given a text query in a certain language. Following [27], we first uniformly sample 12 frames from each video, and use the pre-trained vision encoder to extract representations for each frame. We then perform mean pooling over frame representations to get the video representation. We also evaluate the models under three settings as in Sec.4.3. We report the text→video Recall@1 score in Table 3. Under *Zero-shot* setting, MLA$_{\text{CLIP}}$, which is trained on CC69L without using any video data, achieves comparable or even better results than the fine-tuning results of the state-of-the-art M-VLP model XLM-R-MMP [16] on several languages (de: 20.1 vs. 21.1; fr: 22.0 vs. 21.8; es: 20.2 vs. 21.9). Under the *Fine-tune on English* and *Fine-tune on All* settings, MLA$_{\text{CLIP}}$ also outperforms XLM-R-MMP significantly. We consider the convincing performance comes from two reasons: 1) CLIP is a strong VLP model that can generalize well on video data. 2) The proposed MLA framework can well transfer the open-domain knowledge learned by CLIP to other languages. These results suggest that MLA could maintain the open-domain capability of the VLP model which generalizes well on different downstream data.

Table 3: Multilingual video-text retrieval results on MSRVTT. ZS: *Zero-shot*

| | Method | en | de | fr | cs | zh | ru | vi | sw | es | mean |
|---|---|---|---|---|---|---|---|---|---|---|---|
| ZS | Ours(MLA$_{\text{CLIP}}$ w/o LE) | **30.8** | 18.3 | 18.9 | 14.5 | **18.6** | 12.6 | 7.2 | 10.2 | 19.3 | 16.7 |
| | Ours(MLA$_{\text{CLIP}}$) | **30.8** | **20.1** | **22.0** | **15.7** | 18.3 | **14.4** | **8.2** | **10.7** | **20.2** | **17.8** |
| FT-En | XLM-R-MMP [16] | 23.8 | 19.4 | 20.7 | 19.3 | 18.2 | 19.1 | 8.2 | 8.4 | 20.4 | 17.5 |
| | Ours(MLA$_{\text{CLIP}}$) | **42.5** | **26.1** | **26.7** | **20.5** | **25.3** | 18.9 | **12.9** | **12.6** | **27.2** | **23.6** |
| FT-All | XLM-R-MMP [16] | 23.1 | 21.1 | 21.8 | 20.7 | 20.0 | 20.5 | 10.9 | 14.4 | 21.9 | 19.4 |
| | Ours(MLA$_{\text{CLIP}}$) | **42.5** | **33.1** | **34.5** | **30.5** | **31.6** | **28.9** | **16.9** | **24.3** | **33.5** | **30.6** |

## 4.5 Ablation Studies

### A. Training Strategy

We conduct an ablation study in Table 4 to validate the effectiveness of the proposed MLA training strategy. For those settings with NLT and LE at the same stage, we add the loss of the two objectives together during training. By comparing row 1 to row 2&3, we observe that LE at stage one leads to poor performance. This

Table 4: Ablation study on training strategy.

| Row | Stage one | | Stage two | | Multi30K | | | MSCOCO 1K | |
|---|---|---|---|---|---|---|---|---|---|
| | NLT | LE | NLT | LE | de | fr | cs | ja | zh |
| 1 | ✓ | | | | 76.3 | 74.2 | 67.2 | 72.1 | 75.7 |
| 2 | | ✓ | | | 68.2 | 67.7 | 58.6 | 65.9 | 71.7 |
| 3 | ✓ | ✓ | | | 71.1 | 69.7 | 59.8 | 67.6 | 73.9 |
| 4 | ✓ | | | ✓ | **78.7** | **77.7** | **70.8** | **74.9** | **78.5** |
| 5 | ✓ | | ✓ | ✓ | 78.4 | 77.3 | 69.9 | 74.2 | 78.1 |

indicates that aligning with the native language is more important for the VLP model to acquire new languages at an early stage. It is consistent with the learning habits of humans. By comparing row 1 and row 4, we see that LE at stage two could bring improvements on the new languages. Additionally, comparing row 4 and row 5 suggests that optimizing the model with NLT and LE together at stage two does not bring improvements.

### B. Language Acquirers and Embedding Initialization

In order to validate the effectiveness of the proposed Language Acquirers, we remove the language acquirers and the M-BERT embedding initialization from the model respectively and evaluate on zero-shot multilingual image-text retrieval. As shown in Table 5, the performance on all languages drops significantly without language acquirers. Meanwhile, initializing the embedding with M-BERT [8] only brings incremental improvements. It indicates that the language

Table 5: Ablation study over Language Acquirers (LA) and Embedding Initialization (EI).

| Methods | Multi30K | | | MSCOCO 1K | |
|---|---|---|---|---|---|
| | de | fr | cs | ja | zh |
| MLA$_{\text{CLIP}}$ | **78.7** | **77.7** | **70.8** | **74.9** | **78.5** |
| MLA$_{\text{CLIP}}$ w/o LA | 76.1 | 74.9 | 65.7 | 70.3 | 76.5 |
| MLA$_{\text{CLIP}}$ w/o EI | 77.9 | 76.2 | 69.4 | 74.6 | 78.1 |

acquirers contribute most to the performance, and MLA does not depend much on the initialization of non-native embedding.

## C. Comparing with Machine Translation based method

We compare MLA with a translate-based method CLIP+MT as follows. For non-English text queries in Multi30K and MSCOCO, CLIP+MT first translates them into English with a strong commercial machine translation service Iflytek[5], and then performs image-text retrieval. As shown in Table 6, the CLIP+MT performs significantly worse than MLA in all languages. This is because the translate-based method can introduce extra translation errors or ambiguity although the commercial translation service has been optimized with a large volume of multilingual corpus. By directly optimizing the alignment between vision and each language, MLA does not suffer from this issue.

Table 6: Comparing with the machine translation based baseline.

| Method | Multi30K | | | MSCOCO 1K | |
| --- | --- | --- | --- | --- | --- |
| | de | fr | cs | ja | zh |
| CLIP+MT | 75.3 | 73.0 | 66.3 | 72.2 | 74.2 |
| $MLA_{CLIP}$ | **78.7** | **77.7** | **70.8** | **74.9** | **78.5** |

## D. Low-resource Languages

Image-text pairs may be rare for low-resource languages. To explore the performance of MLA under this situation, we further simulate a **low-resource scenario** using XTD dataset. We fine-tune $MLA_{CLIP}$, $UC^2$ and MURAL with small amount of data from XTD in an unseen language. Both $UC^2$ and MURAL are pre-trained on CC6L with 3M images. We randomly sample 600 pairs for finetuning, and the remained 400 samples are evenly divided for validation and testing. Korean is chosen to perform simulation as its script and language family are not covered by CC6L. Experimental results in Table 7 show that MLA can achieve competitive results with **very small amount of text-text pairs only** (row 3), and adding image-text pairs brings further improvement (row 4). It demonstrates that MLA is still an attractive method for low-resource languages even without any image-text pairs.

Table 7: Low resource performance on image-Korean retrieval.

| Row | Method | Data | Training samples 100 / 200 / 600 |
| --- | --- | --- | --- |
| 1 | $UC^2$ | Img-Txt | 47.0 / 60.1 / 78.3 |
| 2 | MURAL | Both | 53.2 / 60.8 / 73.3 |
| 3 | $MLA_{CLIP}$ | Txt-Txt | 51.7 / 62.8 / 78.7 |
| 4 | $MLA_{CLIP}$ | Both | **56.7 / 66.9 / 80.1** |

## E. Amount of Training Data

We conduct experiments to control the numbers of image-text pairs used for each language. We train the models with CC6L and evaluate on MSCOCO 1K and Multi30K under the zero-shot setting. The corresponding mean AR over non-English languages (de, fr, cs, ja, zh) are drawn in Figure 3. We observe that MLA performs significantly better than MKD [32] in all cases. Note that when the amount of training data is small, the advantage of MLA is more obvious, which could outperform MKD even without the LE training stage. Additionally, when training with only 30K image-text pairs per language, MLA outperforms $UC^2$, which is pre-trained with 3M pairs per language. MLA is thus a data-efficient method to build multilingual and multimodal models.

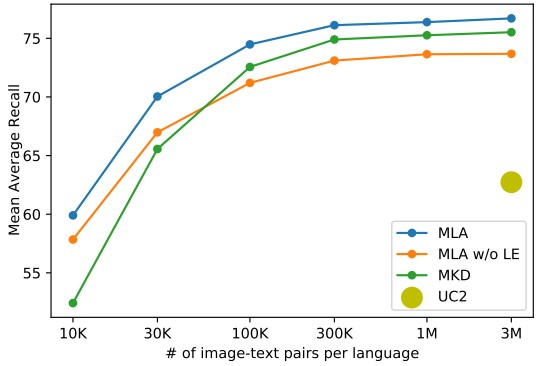

Figure 3: Mean AR vs. Number of data pairs.

## F. Language Extensibility

Multilingual models often encounter the need to support new languages that do not occur in the training stage. We conduct language extension experiments to compare $MLA_{CLIP}$ with M-VLP model $UC^2$ [44] on the XTD dataset [1]. XTD supports 11 languages, and 5 of them (en, de, fr, cs, zh, ja) are seen in the pre-training stage of $UC^2$, while other 6 languages (it, es, ru, pl, tr, ko) are unseen. To make a fair comparison, we first train $MLA_{CLIP}$ with the same data as $UC^2$ and then train both of them on unseen languages with CC69L. The zero-shot image-text retrieval results on XTD are shown in Table 8. We observe a significant performance degeneration on the seen languages for $UC^2$ when training solely with unseen languages (row 1 vs. row 2). Even keep training with the seen languages, the performance is still significantly reduced due to the limited model capacity (row 1 vs.

---

[5]https://global.xfyun.cn

row 3). In contrast, as MLA decoupled multiple languages through acquirers, the performance of the seen languages is rarely affected (row 4 vs. row 5) . This suggests that MLA framework can build multimodal multilingual models that are suitable for supporting increasing numbers of languages.

Table 8: Language extention experiments on XTD dataset.

| Row | Method | Seen languages | | | | | Unseen languages | | | | | |
|-----|--------|------|------|------|------|------|------|------|------|------|------|------|
| | | en | de | fr | zh | ja | it | es | ru | pl | tr | ko |
| 1 | UC$^2$ w/o unseen language training | 71.8 | 67.5 | 68.4 | 61.9 | 51.5 | - | - | - | - | - | |
| 2 | UC$^2$ w/ unseen language training | 63.6 | 57.8 | 57.6 | 57.6 | 48.4 | 56.4 | 56.2 | 51.3 | 56.4 | 51.6 | 51.3 |
| 3 | UC$^2$ w/ all language training | 65.2 | 59.3 | 59.7 | 60.1 | 50.5 | 57.7 | 56.5 | 50.9 | 55.3 | 53.2 | 50.2 |
| 4 | MLA$_{\text{CLIP}}$ w/o unseen language training | 75.9 | **72.6** | **72.9** | 73.7 | **67.2** | - | - | - | - | - | - |
| 5 | MLA$_{\text{CLIP}}$ w/ unseen language training | **76.0** | **72.6** | **72.9** | **73.8** | **67.2** | **64.7** | **62.8** | **58.1** | **63.0** | **56.5** | **57.3** |

## 5 Conclusion and Limitations

In this paper, we propose the Multi-Lingual Acquisition (MLA) framework that can empower multilingual capability on monolingual Vision-Language Pre-training models with low-cost and high-flexibility. MLA injects language acquirers and a non-native embedding block into VLPs to support new languages. Inspired by the language learning habits of humans, we propose a two-stage training strategy to optimize the language acquirers and non-native embedding block. MLA applied on CLIP achieves state-of-the-art performances on multilingual image-text and video-text retrieval benchmarks with much less computing costs and training data. Extensive ablation studies demonstrate that MLA is a flexible, effective, and efficient method to empower multilingual capability on multimodal models.

Though MLA shows high data efficiency that achieves high performance with a small amount of multilingual training data in our experiments, it has the following limitations, which also suggest potential research avenues: i) MLA relies on bilingual corpora, which limits its application on low-resource languages. Meanwhile, state-of-the-art models, such as UC$^2$ and MURAL, also rely on bilingual information. How to achieve comparable performance with monolingual corpora only is still an open problem. ii) MLA is currently limited to coarse-grained retrieval tasks. We will investigate the further application of MLA to fine-grain tasks like VQA in our future work. Furthermore, the majority of our training data is automatically constructed through machine translation, so the ethical prejudice from the machine translation service may potentially affect the behavior of multilingual models produced by MLA. One way to mitigate such concern is to use human annotated or reviewed data for training.

## 6 Acknowledgement

This work was partially supported by the National Key R&D Program of China (No. 2020AAA0108600), the National Natural Science Foundation of China (No. 62072462), and the Large-Scale Pre-Training Program 468 of Beijing Academy of Artificial Intelligence (BAAI).

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
