# OpenReview forum: "Multi-Lingual Acquisition on Multimodal Pre-training for Cross-modal Retrieval"
_NeurIPS.cc/2022/Conference — NeurIPS 2022 Accept_

### Official Review · Reviewer_uTRM · 2022-07-09

**Rating:** 6
**Confidence:** 4
**Soundness:** 2 fair
**Presentation:** 2 fair
**Contribution:** 2 fair

**Summary:**

### After author response
Thanks authors for running additional experiments, including implementing MURAL. I raise the score to 6.
I think there's a merit in adapting monolingual models to multilingual ones, the approach in the paper is good, but it's overselling. I recommend the authors to reword a bit, saying what is reflected in the results. For instance, in the response the authors said MLA doesn't suffer from the curse of multilinguality based on a result of *only* 11 languages. I don't think you can extrapolate that to hundreds of languages.

-----------------

### Summary
The paper proposes a way to adapt English Vision-Language Pre-training (VLP) to other languages via two stages: Native Language Transfer stage and the Language Exposure stage.  In Native Language Transfer state, parallel data is utilized to adapt language encoder to the new languages using language acquisition encoder, a combination of embeddings for new languages and adapter.. In the Language Exposure stage, only Language Acquisition encoder is trained to match with images via contrastive loss. In the text-image retrieval evaluation, MLA shows strong performances compared to other multilingual VLP. The main contribution of the paper is to propose a lightweight approach turning monolingual VLP into multilingual one.


**Questions:**

### Questions:
- For a low resource scenario (Table 6), since MURAL supports 110 languages, what is the performance of MURAL?
- What is the training size of translation data CC6L and CC69L?
- If you train your dual stream model from scratch for 5 languages in Multilingual image-text retrieval experiment, how long does it take? and what would be the final performance?

### Suggestions:
- I think the analogy to how human learning and human learning habit just makes the paper more difficult to read. From technical perspective, you want to retain as many as pre-trained parameters as possible, so you adjust the parameters of new languages. It makes sense. Maybe just say that.

**Limitations:**

Yes.

**Strengths And Weaknesses:**

### Strengths
- A lightweight approach for making monolingual VLP multilingual.
- Evaluation on zero-shot, finetuning English and fine-tuning all shows substantial gains over previous work on multilingual VLP (e.g., MURAL, MKD, …)
- On multilingual video-text retrieval, MLACLIP outperformed XLM-R MMP significantly.


### Weaknesses
- Adapting monolingual language pretrained models into multilingual ones has been studied extensively in the literature. For instance, an efficient method was proposed in [[1](https://arxiv.org/abs/2002.07306)] which uses only 1 gpu. I feel that the novelty aspect is a bit short in adapting monolingual VLP to multilingual one. Nevertheless, this is not a major weakness of the paper.
- My main concern is the comparison with previous work. MLA adapts CLIP-ViT-B-32 and CLIP-ViT-B-16 to 4 languages (de, fr, cs, ja) in multilingual image-text retrieval. While MLA is compared with MURAL, it is worth pointing out that MURAL supports 109 languages. MURAL uses EOBT that consists of translations of 109 language pairs and Alt-Text (image-text pairs) covers 110 languages. Thus comparing MLA to MURAL is somewhat comparing an apple to orange. The curse of mulitlinguality tells us that adding more languages into the mix would degenerate the performance of high resource languages, a well known problem in multilingual machine translation. In my opinion, a fairer baseline would be the MURAL system but it is trained on 5 languages similar to MLA. In line 273, when the authors compare MLA to MURAL, note that MURAL has more parameters but they are mostly the embeddings of the rest 104 languages, which will never be updated during fine-tuning. So having more parameters doesn’t mean that it is easier for fine-tuning.
- On computational cost, Table 2 shows that MLA need 1 V100 GPU and 0.5 day to train. This is rather optimistic since it uses pre-trained CLIP-ViT-B-32. I think the cost of training the CLIP model should be added into the comparison.
- While conceptually MLA can support more languages, as I mentioned earlier, adding more languages could lead to performance drop on high resource languages. So the aspect of language extensibility remains to be questioned. Additionally, MLA requires translation of image captions to train the language acquisition encoder. This is problematic because it depends on machine translation systems, which often support a certain number of languages.

---

> ### Author Response · Authors · 2022-08-02
> **Author Response to Reviewer uTRM**
>
> Thanks for carefully reading and providing valuable feedback. We list our responses to the review comments here:
>
> **1."Adapting ... has been studied extensively ... an efficient method ... in [1]"**
> MLA shares a similar idea with RAMEN[1] that transferring a monolingual model to other languages in a fast and effective way. The major difference is that MLA focuses on transferring the VLP model instead of the text-only model. Concretely, RAMEN adopts the MLM to implicitly align two languages. This strategy is not applicable to a VLP model like CLIP, since it is not pre-trained with MLM. Moreover, directly adopting the objective of CLIP does not perform well (Table 4, row 2). To address this, we design the two-stage training strategy that first aligns target languages to the source language and then to vision. Thanks to the independence of language acquirers, our model can flexibly extend to new languages.
>
> **2."A fairer baseline ... MURAL ... trained on 5 languages "**
>
> To clarify, MLA adapts CLIP into 5 languages (de,fr,cs,ja,zh). Since some baselines do not report the result on COCO-CN, we do not present it in Table 2.
> We understand the concern that MURAL may suffer from the curse of mulitlinguality. However, it is also optimized with more data on the 6 languages. To make a fair comparison, we implement MURAL (not released) on the 6 languages considering our computing budgets. We train MURAL on CC300K(same with MLA) using 1 V100 GPU, and on CC3M (the largest dataset we can access) with 8 V100 GPUs. Both models converge in 1 day and 4 days respectively. We find that initializing the dual-encoder of MURAL with CLIP and M-BERT boosts the performance. Even doing so, both two models perform worse than MLA. Note that under the fair comparison, MLA also shows its low-cost merit, since the data and computing resources of MURAL-CC3M is much larger.
>
> ||en-m30k|de|fr|cs|en-coco|ja|zh|
> |---|---|---|---|---|---|---|---|
> |MURAL-CC300K|67.8|62.7|60.8|57.5|68.1|62.5|67.0|
> |MURAL-CC3M w/o init|73.7|69.2|66.0|61.4|74.5|68.9|73.2|
> |MURAL-CC3M|79.3|73.7|72.4|69.2|76.1|71.1|74.9|
> |MLA-CC300K|84.4|78.7|77.7|70.8|79.4|74.9|78.5|
>
> **3."Line 273 ... MURAL has more parameters ... mostly the embeddings"**
>
> Apart from the embeddings, the text encoder of MURAL is larger than MLA (768-d, 12 heads vs 512-d, 8 heads). We will make the statement here more precise.
>
> **4."Cost of training CLIP should be added ..."**
>
> Table 2 shows the cost of multilingual and multimodal training. Note that most M-VLP models are based on monomodal pre-training e.g. XLM-R. They typically do not consider it when reporting the computational cost.
> Additionally, CLIP14 is trained for 12 days on 256 V100. It is much lower than MURAL Large, which is trained on 512 TPUv3 for 17 days. We cannot report the training costs of CLIP32 and CLIP16 since they are not mentioned in the paper.
>
> **5."Adding more languages ... performance drop ... "**
>
> We notice the curse of multilinguality issue. However, it occurs in the current M-VLP model but not in MLA. Table 7 suggests that after incorporating unseen languages, UC2 suffers from performance decreases on seen languages, but MLA doesn't since MLA handles the different languages with independent language acquirers.
>
> **6."MLA requires translation of image captions to train ... "**
>
> We agree that a limitation of MLA is the reliance on bilingual corpora. So far, state-of-the-art models, such as UC2 and MURAL, all rely on bilingual information. MLA reaches comparable performance but uses fewer bilingual pairs compared with these works. M3P explores relying on monolingual corpora only, but it achieves sub-optimal performance. We will further investigate how to achieve comparable performance without bilingual information in multilingual image-text retrieval in future work.
>
> **7."For 'MURAL performance in Tabel 6'"**
>
> We cannot access the official 110 languages MURAL. As a remedy, we conduct the low resource experiment on MURAL-CC3M:
> |# of training samples|100|200|600|
> |---|---|---|---|
> |UC2|47.0|60.1|78.3|
> |MURAL-CC3M|53.2|60.8|73.3|
> |MLA|56.7|66.9|80.1|
>
> **8."What is the training size of translation data CC6L and CC69L?"**
>
> CC6L contains 3M sentences for each of the 6 languages. Since we remove texts with broken image urls, we use about 2.5M. MLA in Table 1 is trained for randomly sampled 300K images from CC6L. The subset can be found in our code (see README.md). As for CC69L, it contains 20K captions in 69 languages.
>
> **9."If you train your model from scratch ... "**
>
> The basic idea of MLA is to empower an existing monolingual model with multilingual capability. There is no need to train the model from scratch. As mentioned above, we conduct a fairer comparison between MURAL and MLA by training MURAL-CC3M. This comparison indicates that MLA requires less computing and data resources.
>
> **10."The analogy to how human learning ... "**
>
> Thank for the suggestion about paper writing. We will consider it to prepare our updated version.

---

### Official Review · Reviewer_9Y6f · 2022-07-10

**Rating:** 7
**Confidence:** 4
**Soundness:** 3 good
**Presentation:** 3 good
**Contribution:** 3 good

**Summary:**

The paper proposes a Multilingual Acquisition (MLA) framework that leverages vision and language modality at a low cost. MLA empowers monolingual VLPs with multilingual capability using much less training data through a language acquisition encoder. The authors apply the proposed method to multilingual image-text and video-text retrieval benchmarks with much less training data and parameters. In the training process, they present a two-stage training strategy to train the language acquisition encoder: 1) Native Language Transfer (NLT) stage, the model is trained to learn the connection between the native and new language, then followed by 2) Language Exposure (LE) stage, the model to learn the alignment between new languages and images.

**Questions:**

- I was wondering why did you only choose English as the source language for the zero-shot setting other than English is the high-resource language. Have you investigated the zero-shot setting where English is not the source/pivot language? It would be interesting to find out whether other languages? Many of these evaluated languages are not closely related to English (e.g., Japanese, Chinese).

**Limitations:**

Yes, the authors address the limitations.

**Strengths And Weaknesses:**

Strengths:
- A simple two-step approach to acquire a new language by leveraging monolingual VLP by aligning the languages and modalities. The approach is straightforward and effective.
- This is a solid paper; it has comprehensive results and analysis
- The proposed approach has strong zero-shot and few-shot capability,

Weaknesses:
- Didn't find any substantial weakness to report.

Typo:
- MultiLingual => Multilingual. It is not common to write multilingual with a capital L. Another way to write it is Multi-lingual.

---

> ### Author Response · Authors · 2022-08-02
> **Author Response to Reviewer 9Y6f**
>
> Thank you for spending time reading our paper and throwing interesting questions. We make responses to your concerns as below:
>
> **1. "It is not common to write multilingual with a capital L ... "**
>
> Thanks for pointing out this, we will fix it in the updated version.
>
> **2. "Why did you only choose English as the source language ..."**
>
> Currently, we have not investigated transferring from non-English languages. We choose English as the only source language since we cannot find any counterpart non-English models with CLIP. However, it is technically feasible to apply MLA to non-English VLP models. The choice of source languages is likely to have an impact on the target language performance. This phenomenon has been observed in cross-lingual NLP tasks such as NLI  [https://arxiv.org/abs/1909.00964]. We will investigate it under the cross-modal scenario in our future work.

---

> > ### Comment · Reviewer_9Y6f · 2022-08-09
> > **Thank you for the response.**
> >
> > Thank you for answering. I would say this is excellent work. All the best for your submission!

---

### Official Review · Reviewer_spgr · 2022-07-11

**Rating:** 5
**Confidence:** 4
**Soundness:** 2 fair
**Presentation:** 2 fair
**Contribution:** 2 fair

**Summary:**

The paper try to solve the problem that existing vision-language pre-trained models require huge computing resources and cannot be flexibly extended to new languages. They mainly propose a two-stage training strategy to optimize the language acquisition encoder, namely the Native Language Transfer stage and the Language Exposure stage. The authors compare their model to different baselines on multilingual image-text and video-text retrieval benchmarks, and find that their model compares well to the current state-of-the-art.


**Questions:**

1.	Why use the average score over Recall@1, Recall@5, and Recall@10 as your evaluations? What's the idea behind it, and not Recall@1, Recall@5, and Recall@10, or something else?
2.	I want to know whether the Language Acquier is shared in different languages? This is very important to evaluate the size of parameters of model.
3.	I want to know the training strategy. Whether different languages are trained in parallel or serial ways? For a language, whether the model only is trained on this language? And, the model have not the multilingual ability?
4.	The comparison of Table 2 is very unfair. The baselines are on trained on many different languages at the same time. However, the model of this paper is only trained on a language for some lingual applications and the baselines are suitable for different languages. Even though the model are trained on some languages as the papers, the trained number of languages is less than M3P.
5.	The description of the model in Section 3 needs a lot of improvement. It is confusing and many times convoluted. There are too many variables being introduced.
6.	It is not clear whether you do not introduce the equations for the enumerated items, or whether you do not use the enumerated items in your model architecture. Please clarify.
7.	The paper writing is poor. I found a lot of grammar mistakes in the paper which need to be corrected.


**Limitations:**

yes

**Strengths And Weaknesses:**

The paper main issue is the lack of clarity at certain crucial points, especially in Section 3 (the model description). The maths is at times too convoluted and should be simplified and improved. Moreover, the major concern I have about this paper is the experimental results. The work is relatively novel, although a deeper discussion of why the specific architecture used was not done and would have been important to
understand the contributions.

---

> ### Author Response · Authors · 2022-08-02
> **Author Response to Reviewer spgr**
>
> Thank you for spending time to read our paper and providing valuable feedback. We clarify some of the questions in the review here:
>
> **1. "For 'Strengths And Weaknesses'"**
>
> This paper proposes MLA, a simple-yet-efficient approach to construct a multilingual model that can easily extend to other languages. It requires much fewer training resources (e.g. computing resources and multilingual data) and achieves better performance compared with existing pre-training-based methods. Comprehensive ablation studies verify the effectiveness of the proposed training strategy and components of MLA. We will keep polishing the paper to make it more friendly to readers.
>
> **2. "Why use the average score over Recall@1, Recall@5, and Recall@10 ... "**
>
> Recall@1, Recall@5, and Recall@10 are common metrics for retrieval tasks. For multilingual image-text retrieval experiments, reporting Recall@1, Recall@5, and Recall@10 of each language makes the table quite redundant. Therefore, for the sake of brevity,  we show the average score, the same as M3P, UC2, and MURAL do.
>
> **3. "Whether the Language Acquier is shared ..."**
>
> As mentioned in the paper (line 165),  the language acquirer is not shared in different languages. It is worth mentioning that the majority of the parameters in the MLA framework come from the embedding matrix (92.2M, line 237), which is shared across all languages. Since the size of a single language acquirer is relatively small (3.14M for each language, line 235), supporting 100 languages like M3P will cost about 407 M parameters (3.14x100 + 92.2≈407M) in total, which is still less than M3P (566M).
>
> **4. "Whether different languages are trained in parallel or serial ways? ... model have not the multilingual ability? "**
>
> All languages are trained in parallel, but each language is processed with its individual language acquirers. We argue that the MLA framework has multilingual ability. In a plug-to-play manner,  it can transfer a monolingual VLP model into a multilingual one that can handle any type and any number of languages. This design has two major advantages: low data consumption and strong language extensibility. Experimental results in Section 4.5.C and 4.5.D suggest that MLA can handle a new language with a little amount of training data, and Section 4.5.E indicates that extending to new languages with MLA does little harm to existing languages.
>
> **5. "Comparison of Table 2 is very unfair. The baselines are on trained on many different languages"**
>
> It is worth noting that our model supports the same languages as UC2, so at least the comparison between MLA and UC2 is fair in Table 2. We will add a column in Table 2 to indicate the number of supported languages.
>
> **6. "Section 3 needs a lot of improvement ... whether you do not introduce the equations for the enumerated items ... "**
>
> Thank you for the suggestions. We will keep polishing the paper. For brevity, we omit the enumerated items of variables $S$, $T$, and $V$ in the equations and the model illustration.

---

### Official Review · Reviewer_4qbz · 2022-07-11

**Rating:** 7
**Confidence:** 4
**Soundness:** 3 good
**Presentation:** 3 good
**Contribution:** 2 fair

**Summary:**

___________AFTER THE RESPONSE______________

I would like to sincerely thank the authors for their additional experimental efforts, aiming to provide coherent responses to all my questions and suggestions. Including such experiments into the paper (and adding relevant discussions) would make the paper much stronger. If the focus of the work is on retrieval tasks, I strongly suggest the authors to reflect that fact in the title and also in the introduction, and add discussions focused on why they don't extent their framework to other tasks. The title and the introduction of the paper should be changed to reflect this and avoid 'overselling'. The paper should also make its connections to previous work on adapters more apparent, and clearly extract its core novelty in comparison to other related work in this field (e.g., such as the adapter-based approaches from the xGQA paper). I raise my score from 6 to 7 to acknowledge the wealth of extra experiments conducted for the response, and a very detailed response. However, this increase (as a note to the area chair as well) is conditioned on really incorporating all the promised changes and edits to the final camera-ready paper, if the paper gets accepted.
++++++++++++++++++++++++++++++++++++++++++++


This paper proposes a novel framework for multilingual multimodal learning that: 1) bypasses the need for joint and very expensive multilingual multimodal pretraining (e.g., M3P); and 2) adapts to multilingual setups to parameter-efficient modules (called language acquirers in this work, where these acquirers are basically standard bottleneck adapters). Besides more cost-friendly training of the model, the new framework, termed MultiLingual Acquisition (MLA)*, yields strong results in several cross-modal video-text and image-text retrieval tasks; the underlying model is a standard dual-strema model: CLIP-ViT-B, which is already through pretraining design bound to excel in (monolingual) retrieval tasks. The paper also presents a series of ablation studies and side experiments, covering important aspects such as training strategy, embedding initialisation, simulating low-resource languages and scenarios, zero-shot evaluations (on unseen languages) and amount of training data. There are additional qualitative and quantitative analyses available in the appendix.

*Minor: I would suggest a more precise name for the framework (that also includes multi-modality somewhere)

**Questions:**

- This is not a criticism but more a suggestion: conducting some experiments with other underlying models beyond CLIP (and/or CLIP-style models of different sizes) would offer other useful comparisons and points of reference for future work.

- It would also be interested to try out language acquirers with pretrained M-VLP models like M3P. Can MLA help M3P to adapt/specialise for a particular source/target language? This would be a very useful addition to the paper.

- Minor: I suggest to cite papers based on their publications in conference proceedings rather than arxiv versions.

**Limitations:**

The statement in limitations of learning 'coarse-grained' versus 'fine-grained' representations and limitations seems quite hand-wavy to me - what is meant by this exactly? I would also suggest to be more open with other limitations of the work (e.g., some of them are mentioned in this review - (i) there weren't any real low-resource languages in evaluation; those scenarios were mostly simulated; (ii) application and evaluation is currently tied only to retrieval tasks; (iii) there's no discussion of potential avenues for future work, etc.)

**Strengths And Weaknesses:**

Strengths:
=========
- The motivation for the work is very clean and clearly stated. The core methodology is soundly described, and the paper is very easy to follow in general.
- The results in the standard cross-modal retrieval tasks are consistently positive.
- The ablation study and additional experiments are also very well motivated and very comprehensive, and offer the readers many interesting analyses and points of discussion.
- The methodology, along with the core hypotheses, is very sound, and the experimental setup aligns well with the methodology.

Weaknesses:
==========
- The current MLA framework is tested and seems to work only on retrieval tasks. A more comprehensive analysis on other, structurall different multilingual multi-modal tasks would make the main findings (and consequently, the impact of the work) much stronger. I strongly suggest to extend the paper to more underlying models (also single-stream models), and evaluate on a carefully designed recent IGLUE benchmark https://arxiv.org/pdf/2201.11732.pdf). The paper should convince the reader that MLA is not intended only for retrieval tasks (or the other way around - if it doesn't work as well for other tasks, to narrow the scope of the work to retrieval only in the title and in the introduction).
- One of the core components of the work are 'language acquirers': they are basically standard language adapters, examined to detail in the body of work on cross-lingual text-only transfer (e.g., see again the cited MAD-X work from Pfeiffer et al.). Having said that, the idea of 'language acquirers' is basically also very related to another work of Pfeiffer et al. (cited in the paper as "[28] Jonas Pfeiffer, Gregor Geigle, Aishwarya Kamath, Jan-Martin O Steitz, Stefan Roth, Ivan Vulić, and Iryna Gurevych. xgqa: Cross-lingual visual question answering. arXiv e-prints, 2021."). More discussion is needed here so that the reader really understands the connections to language adapters and connections to the adaptation work done on multilingual multi-modal work in previous work. In my understanding, the idea is basically very similar, with two core differences: a) MLA is applied on a dual-stream model which is suitable for retrieval tasks but might not be suitable for e.g. VQA tasks; b) Pfeiffer et al., relied on MLM-ing for adaptation, while MLA relies on specific bilingual information to learn transformations from the target language to the language of the monolingual VLP model (i.e., English).
- There is no comparison with standard translation-based baselines. A very basic question: why would one want to go through the MLA procedure if translation-based approaches can work better? Some comparison numbers must be provided.
- The chosen retrieval datasets might paint a wrong picture about the difficulty of the task: I would suggest the authors again to check the IGLUE benchmark and also to evaluate on the recently proposed WIT dataset. IGLUE also covers some lower-resource languages and its tasks are generally more difficult than the standard overly optimistic evaluations on Multi30K, and MSCOCO-1k and 5k.

---

> ### Author Response · Authors · 2022-08-02
> **Author Response to Reviewer 4qbz**
>
> Thank you very much for your careful reading and valuable feedback. Below we list our responses to your concerns:
>
> **1. "MLA framework is tested... only on retrieval tasks ... "**
>
> This paper aims to empower a monolingual VLP model to a multilingual model. Since most existing works on multilingual and multimodal learning evaluate their models on cross-modal retrieval, we also focus on these tasks to make a comparison with them. We will consider the suggestion to limit the scope to retrieval for this paper. However, we believe that MLA is also appliable for single-stream models. A straightforward approach is to perform NLT and LE at the [SOS] token, whose hidden state is used for image-text retrieval and VQA for the single-stream models. We will investigate it in our future work.
>
> **2. "More discussion is needed ... the connections to language adapters... "**
>
> Thanks for the suggestions, we will add more discussions on the differences between MLA and MAD-X, xGQA in the Related Work.
>
> **3. "There is no ... translation-based baselines."**
>
> We compare MLA with a translate-based baseline CLIP+MT as follows. For non-English queries, CLIP+MT first translates them into English with a strong commercial machine translation service Iflytek (https://global.xfyun.cn), and then performs image-text retrieval.
>
> Multi30k and MSCOCO image-text retrieval. Metric: AR
> | Model|de|fr|cs|ja|zh|
> |---|:---:|:---:|:---: |:---:|:---:|
> | CLIP+MT | 75.3 | 73.0 | 66.3 | 72.2 | 74.2 |
> | MLA| **78.7** | **77.7** | **70.8** | **74.9** | **78.5** |
>
> As shown in the above table, the CLIP+MT performs significantly worse than MLA in all languages. This is because the translate-based baseline can introduce extra translation errors or ambiguity although the commercial translation service has been optimized with a large volume of multilingual corpus. By directly optimizing the alignment between vision and each language, MLA does not suffer from this issue.
>
> **4. "The chosen retrieval datasets might paint a wrong picture about the difficulty of the task...Check the IGLUE benchmark"**
>
> For fair comparisons with existing works, we choose Multi30K and MSCOCO as our main benchmarks. Following your suggestion, we further evaluate MLA and MURAL-CC3M on WIT. The results are shown in the following tables.
>
> WIT Text-to-Image retrieval. Metric: Recall@1
> ||ar|bg|da|el|et|id|ja|ko|tr|vi|avg|
> |---|:---:|:---:|:---:|:---:|:---:|:---:|:---:|:---:|:---:|:---:|:---:|
> |MURAL-CC3M|13.4|12.2|12.7|15.7|3.0|9.7|2.7|8.6|13.7|15.0|10.7|
> |MLA|**15.5**|**15.1**|**17.6**|**16.1**|**3.6**|**12.2**|2.7|**10.6**|**17.4**|**19.8**|**13.0**|
>
> WIT image-to-Text retrieval. Metric: Recall@1
> ||ar|bg|da|el|et|id|ja|ko|tr|vi|avg|
> |---|:---:|:---:|:---:|:---:|:---:|:---:|:---:|:---:|:---:|:---:|:---:|
> |MURAL-CC3M| 12.7| **13.3** | 12.9| **15.1** | 9.0| 14.7|13.4 | **10.1**| **14.6** | 16.9|13.3|
> |MLA| **15.3**|12.9  | **15.7** | 14.6 | **9.7** | **17.0** | **13.5** | 9.8 | 12.2 | **18.0** | **13.9** |
>
> For a fair comparison, both MLA and MURAL-CC3M are trained on CC69L. Note that the MURAL-CC3M here is pre-trained on CC3M with CLIP and M-BERT initialization since the original MURAL is not released. More training details of MURAL-CC3M can be found in *our response to Reviewer uTRM*. As shown in the tables, WIT is indeed much harder than Multi30K and MSCOCO since the models achieve lower scores. Besides, MLA still outperforms MURAL-CC3M in most languages on this benchmark, which validates the effectiveness of our method.
>
> **5. "Conducting some experiments with other underlying models beyond CLIP ..."**
>
> Following your suggestion, we additionally apply MLA to CLIP in different sizes with two kinds of architectures: ResNet(RN) and ViT. The results shown below are consistent with the statement in the paper that MLA can perform better on all languages when stronger monolingual VLP is provided. We will add these results to the appendix.
>
> ||m30k-en|m30k-de|m30k-fr|m30k-cs|coco-en|coco-ja|coco-zh|
> |---|:---:|:---:|:---:|:---:|:---:|:---:|:---:|
> |RN50|84.2|76.6|75.8|67.5|78.3|72.7|75.9|
> |RN101|83.9|76.9|77.3|70.4|78.9|73.1|76.9|
> |RN50x4|86.0|80.7|80.3|73.1|80.4|75.5|78.2|
> |RN50x16|87.8|80.6|79.9|73.8|81.7|74.4|77.6|
> |RN50x64|89.9|84.2|84.1|78.1|82.2|79.3|80.6|
> |ViT-B-32|84.4|78.7|77.7|70.8|79.4|74.9|78.5|
> |ViT-B-16|86.4|80.8|80.9|72.9|80.9|76.7|79.2|
> |ViT-L-14|87.9|83.1|83.5|77.0|82.5|78.5|79.1|
>
> **6. "Can MLA help M3P ... for a particular source/target language?"**
>
> As mentioned above, it is technically feasible to apply MLA on a single-stream model like M3P to specialize in a particular language. We will mention this in the Conclusion and Limitation section as a potential avenue for future work.
>
> **7. "Cite papers based on their publications ... more open with other limitations ... a precise name ...  "**
>
> Thanks for the suggestion. We will carefully check the reference list and fix them in the updated version. We will also refine the discussion of Limitations accordingly.

---

### Meta-Review · Area_Chair_jbU6 · 2022-08-26

**Recommendation:** Accept
**Confidence:** Less certain

**Metareview:**

All reviewers are positive to this paper. The authors also respond actively to address the problems raised by the reviewers. I recommend acceptance. One thing is that reviewer 4qbz has pointed out the authors should be responsible to take the comments into consideration and
do necessary changes in the paper, such as adjusting the title and the introduction and adding connections to previous work.

**Award:**

No

---

### Decision · Program_Chairs · 2022-09-14

Accept